# The Influence of Relational Capital on the Sustainability Risk: Findings from Chinese Non-State-Owned Manufacturing Enterprises

**Dongsheng Zhang, Hongwei Wang *** 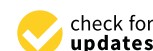 **and Wenfu Wang**

School of Economics and Management, Hebei University of Technology, Tianjin 300401, China; zdsheng@sina.com (D.Z.); study212@163.com (W.W.)
* Correspondence: wanghongwei0714@163.com; Tel.: +86-15822727375

**Abstract:** During the COVID-19 pandemic, the global economy fluctuated while the Chinese economy remained relatively stable—a distinction that has aroused people's curiosity about the unique operation of Chinese enterprises. Compared with the regularity and competitiveness of traditional market strategy theory, Chinese business management pays more attention to informal institutions and relational capital, which is one of the key features that distinguishes Chinese firms from their Western counterparts. Yet, theoretical research on relational capital against the Chinese cultural background remains scarce, and the particularity laws of the socialist market economy are still unclear. Based on the social capital theory, this paper redefines the concept of relational capital in the context of China and uses factor analysis to construct a relational capital measurement index. On this basis, non-state-owned manufacturing enterprises are then used as a sample to explore the interactive relationship between relational capital and sustainability risk. The empirical results show that relational capital can effectively reduce sustainability risk and ensure sustainable operation. In addition, enterprise growth, enterprise development, and marketization can strengthen the role of relational capital and positively regulate the relationship between relational capital and sustainability risk. This paper innovatively constructs the concept and index system of relational capital in the Chinese context, which is the perfection of relational capital theory. At the same time, it verifies the impact of relational capital on business sustainability, revises the correct cognition of relational capital, and supplements the deficiencies of the extant social socialist market economy research. Supported by both theoretical research and empirical conclusions, corresponding management suggestions are put forward for enterprises, governments, and managers to scientifically guide management practice and provide new ideas for future Chinese-style economic research.

**Keywords:** relational capital; sustainability risk; social capital theory; factor analysis method

## 1. Introduction

During the COVID-19 pandemic, the global economy experienced significant turbulence, even as the Chinese economy remained relatively stable. The latter's ability to weather this storm, while other economies floundered, has aroused people's curiosity about the unique operation of Chinese enterprises. Compared with the mature market environment, the uniqueness of Chinese enterprises is the role of traditional Confucian philosophy, which emphasizes personal relationships and permeates daily activities and the deep-rooted relationship culture, which influences enterprise [1]. The market system emphasizes regularity and consistency, whereas a relationship culture emphasizes human interactions; thus, they represent two opposite yet interrelated systems that operate outside the enterprise [2]. In China, the relational capital derived from relationship culture, as an unofficial corporate strategic resource, has become an important supplement to the formal system [3]. In terms of actual operation, when smooth communication and trust in the arrangement of formal institutions are lacking, Chinese enterprises can coordinate

with informal institutions—represented by relational capital—to promote mutual trust and reduce transaction costs.

With the intensification of market competition and the improvement of non-market theories, scholars have gradually acknowledged the auxiliary impact of relational capital on enterprise management while affirming market competition and strategy's core role. Up to now, scholars have explained the essence and connotation of relational capital from different perspectives such as relational theory, resource view, and capability view, but an authoritative and unified definition has not yet been formed [4]. Due to the relatively nascent state of this field, the overall depth and breadth of the literature remain insufficient, the mechanism of relational capital has not been clearly explained so far, and the consensus theoretical research is insufficient [5]. Roughly, relational capital is the sum of actual or potential resources embedded in, derived from, and emerging in external relational networks [6]. Relational capital brings assets through the creation and utilization of relationships, which make relationships become resources to achieve individual and collective goals [7]. Most of the research literature on relational capital is placed in specific organizational relationships, such as supply chains and corporate alliances, and less in a specific national environment. For example, Cousins et al. took the supply chain as the research object and found that the establishment of relational capital can enhance the deepening of the cooperative relationship among enterprises and then promote the coordinated development of the supply chain [6]. Theoretically, there is a research gap in the in-depth exploration of China's relational capital [8]. As a developing theory, the future research of relational capital still needs to be improved continuously.

Further, Capello and Faggian pointed out that relational capital is influenced by regional culture and may have different forms and manifestations in different regions, and even organizational structure and strategic orientation will have a significant impact on relational capital [9]. To explore the theory and application of relational capital in the Chinese context, it is necessary to start from China's specific economic and cultural background and form its theoretical framework of relational capital. Thus, based on social capital theory, the current research integrates the existing theoretical results, defines and measures the concept of relational capital in the context of China, and takes non-state-owned manufacturing enterprises as a sample to explore the impacts of relational capital on sustainability risk. In addition, it deeply analyzes the regulatory effects of enterprise growth, enterprise development, and marketization in this context. Our work reveals the relationship and internal mechanisms operating between Chinese enterprise relational capital and sustainability risk in an environment of economic system transformation, intending to enrich the literature on relational capital and deeply explore the characteristics of enterprise growth under the socialist market economy.

## 2. Theoretical Basis and Hypotheses

### 2.1. Theoretical Basis

2.1.1. Social Capital Theory and Relational Capital

In the 1960s, management scientists broke through the bottleneck of economic research and for the first time brought interactions between people into the category of economic capital. In this way, the social capital theory took shape. Relationships and relational capital have gradually become the focus of social capital theory, where they are regarded as powerful strategic supplementary tools to help companies acquire resources and enhance their competitive advantages in non-market areas. In the 1980s, Granovetter first publicly proposed the concept of relational capital, defining it as a resource that is potentially embedded in individual and organizational interactions [10]. Subsequently, Morgan conducted more systematic studies of relational capital, pointing out that relational capital is a non-market resource that exists outside the enterprise and can bring value to the enterprise [11].

Although research on relational capital has been conducted for more than 40 years, no consensus has been reached on its connotations [12]. One of the main reasons is that

scholars have not clearly defined the concept of relational capital. The narrow school, represented by Kale and Singh, asserts that relational capital exists only in alliance enterprises, reflecting the degree of mutual trust and interaction between enterprises, and is not prominent in business operations and marketing [13]. In contrast, the broad school, represented by Li and Agostini, proposes that relational capital exists in the whole enterprise and that interactions of enterprises produce relational capital [14,15]. In this view, relational capital is an indispensable resource that can create profits and that incorporates government–enterprise relationships. The ambiguous definitions of relational capital affect the formulation of judgment criteria, resulting in contradictory studies on the effect of relational capital. For example, Granovetter et al. suggested that relationships will cause duplication and redundancy of information resources, which will lead to the enterprise's over-reliance on the relational network and reduce its innovation ability, so they have a negative attitude toward relational capital [10]. Inkpen and Tsang, however, have pointed out that enterprises can promote the circulation and sharing of knowledge and strengthen cooperation through the establishment of relationship networks [16]; thus, these authors support enterprises' investment in relational capital.

In addition, since consensus on the concept of relational capital remains elusive, it is difficult to definitively state how relational capital ought to be measured. Scholars usually define and measure relational capital based on the needs of the research topic and content and based on a specific environmental background [17]. Initially, Sambasivan et al. measured relational capital as trust and commitment among alliance firms and quantified relationship strength using a scale; this approach limited relational capital to strategic alliances [18]. Most of the current measurements of relational capital strength draw on Sambasivan's research, albeit with different degrees of correction and supplementation [19]. For example, Bao added dedicated investment to measure relational capital based on trust and commitment [20], while Hammervoll increased the variable of psychological commitment, such as satisfaction, to expand its scope of application [21]. Navarro et al. measured relational capital by assessing the individual context, management, and teamwork [22]. In contrast, Kotabe et al. put more emphasis on interactivity and determined the strength of relational capital by measuring the intimacy, reciprocity, and communication between cooperative enterprises [23]. These measurement methods are scientific and effective in solving specific research problems. Since they are based on Western corporate culture and theoretical foundations, however, they are not suitable for answering relational capital questions in the context of Chinese companies.

2.1.2. Sustainability Risk and Relational Capital

Most of the mainstream relational capital research has adopted a narrow definition, which views relational capital as the resource advantage generated by interactive activities between supply chain enterprises or alliance enterprises [6]. This definition emphasizes an orderly and free external environment and is more suitable for mature markets. In their transitional economic environment, Chinese enterprises must cope with complex external relations and imperfect market mechanisms. To obtain a comprehensive perspective of their operation, one must consider more than just affiliated enterprises. In addition, most of the existing research mixes government–enterprise connections with relational capital, which blurs the premise of equal exchanges with relational capital [24]. In China, there is inevitably an unequal relationship between government entities and business enterprises. Generally, the government supervises and manages enterprise activities, so the government–enterprise connection is more political [25]. Combining the Chinese context and social capital theory, this study defines relational capital as interactions between companies and external entities of equal status that are intended to help the company attain a central position within the network and enhance its access to information. This definition both highlights the prerequisites for equality and emphasizes the management linkages and business impacts. Notably, the political connection arises from the government–enterprise

communication and is not a simple construction of relational capital; thus, it should be seen as a political strategy, especially for Chinese enterprises.

Existing research on the effects of relational capital has mainly focused on resource sharing or cooperative innovation [26,27], with few scholars having explored the relationship between relational capital and sustainability risk. Sustainability can be considered as the degree to which present decisions of organizations impact the future situation of the natural environment, societies, and business viability [28]. From a corporate perspective, sustainability is how a firm grows and develops its recognition of environmental, social, and economic issues and encourages businesses to frame decisions in financial, environmental, social, and human effects, to ensure resilience and value creation [29]. In Dyllick and Hockerts' terms, this means corporations must meet their current objectives without diminishing their ability to meet the needs and demands of future stakeholders [30]. Risk refers to the possibility of certain unfavorable elements arising from organizational structure and commitments, resulting in uncertainty and harm [31]. Due to the multifaceted nature of sustainability, sustainability risk is viewed as a comprehensive consideration of environmental risks, economic risks, and social risks [25]. Among them, Anderson defines sustainability risk as potential financial or operational crises in corporate growth arising from liability litigation, consumer boycotts, shareholder action, and international pressure [32].

Sustainability risk roughly involves the environment, economy, and society and is itself a complex concept [33]. In this study, we focus on the analysis of sustainability risk in corporate operations and adopt Altman's research to define sustainability risk as the risk of bankruptcy and unsustainable operations that may occur when companies operate following established strategic guidelines and management forms [34]. It is mostly used to judge the stability and sustainability of the enterprise's long-term operations. Compared with market strategy measures, relational capital requires long-term investment, and its effects need to be felt for a long time. Therefore, the impact of relational capital on enterprise management can be more scientifically and intuitively reflected through the exploration of its effects on sustainability risk [35].

Compared with Western countries, relational capital has a more prominent impact on the operation of Chinese enterprises and sometimes even directly determines the success or failure of their strategy implementation. The mature social capital theory recognizes the importance of relational capital but considers it to be just a cooperative strategy to assist the market strategy [13]. In contrast, for Chinese enterprises, relational capital is more important than market strategy in some cases, though both are mutually complementary equality strategies. To date, however, few theoretical studies on relational capital have been conducted in the context of Chinese enterprises, mainly for two reasons. First, in the overall management research environment, theoretical research emphasizes market behavior, pays attention to rational institutional constraints and market competition, and ignores the impacts of relational capital on business operations. Second, the growth of Chinese enterprises has some unique aspects, and enterprise behavior is influenced by both modern market thought and traditional Confucianism culture. Thus, explorations of this setting must not only emphasize institutionalization but also focus on relations, which are rarely studied. Therefore, it is necessary to improve the research on the role of relational capital in the Chinese context.

*2.2. Hypotheses*

2.2.1. Relational Capital and Sustainability Risk

According to social capital theory, enterprises can improve resource acquisition efficiency and transformation capabilities by building relationship networks [36]. Paul et al. confirmed that relational capital can achieve information sharing, thereby improving work efficiency and reducing production costs [6]. In addition, as an informal system under kinship culture, relational capital can alleviate the pressure caused by imperfect market development, negative policy impacts, and strong government intervention, to a certain

extent, and enhance the firm's ability to resist management risks, thereby stabilizing its sustainable management [25]. Especially in an environment where China emphasizes informal institutional arrangements, relational capital can often bring important competitive advantages to enterprises, such as the latest policy developments or preferential resource conditions, which can effectively resist sustainability risk [37]. Moreover, Huang et al. proved that relational capital can help firms determine market positioning and maintain organizational stability [38]. In the existing research, the conclusion that enterprises can improve competitiveness through enhancing relational capital has been recognized by academia. In Facia's research, it is found that relational capital plays a protective role in supervising illegal activities and stabilizing market development, which can reduce the pressure of supervision and improve the competitiveness of enterprises [39]. Additionally, the research of Roehrich et al. also pointed out that business conditions will directly affect the stability and sustainability of enterprises [35]. In summary, we propose that relational capital can effectively reduce the sustainability risk, to ensure orderly operation.

**Hypothesis 1 (H1).** *Relational capital has a significant negative impact on sustainability risk.*

### 2.2.2. Enterprise Growth, Relational Capital, and Sustainability Risk

According to agency theory, financial data indicate the level of the enterprise's internal control and reflect its growth, which directly affects the decisions made by managers and the formulation of the enterprise's strategy [40]. Williamson integrated agency theory and corporate finance theory and pointed out that the choice of corporate governance is not only affected by transaction costs but also depended on the governance structure and financial reflection [41]. The enterprise can improve its growth ability by optimizing the allocation of resources to achieve sustainable development. Research by Capello and Faggian showed that relational capital does not play a consistent role in different firm characteristics, and the effect of relational resources is closely related to financial performance [9,42]. Therefore, the same strategic measures, when applied to enterprises in different growth states, will yield significantly different strategic effects. Fazzari et al. took American enterprises as samples to test the relationship between corporate growth and management performance and confirmed that enterprises with excellent growth do indeed have better corporate investment performance [43]. Similarly, Hayashi et al. conducted experiments with Japanese companies as research objects and came to the same conclusion [44]. From this, it can be deduced that the better the growth of the enterprise and the more reasonable its capital structure are, the more obvious the effect of relational capital and the stronger its ability to resist sustainability risk will be [45]. This leads to our hypothesis:

**Hypothesis 2 (H2).** *Enterprise growth has a positive moderating effect on relational capital and sustainability risk.*

### 2.2.3. Enterprise Development, Relational Capital, and Sustainability Risk

The establishment of relational capital is a long-term, complex management activity. Different from technological innovation investment and social responsibility strategy, relational capital activities are more inclined to immediacy and discontinuity, so the requirements for current assets are higher [46]. According to capital structure theory, the stock of current assets and short-term solvency reflect the enterprise's state of development, and strong enterprise development suggests greater liquidity, which can promote the implementation of the enterprise's strategies [47]. Agiomirgianakis et al. conducted a regression analysis on Greek manufacturing data, and the results showed that the efficiency of capital structure and capital management affected the company's return rate [48]. In capital structure management, asset liquidity affects the enterprise's ability to pay its debts and respond to strategic changes, which together guarantee its sustainable operation. Dittmar et al. studied the data of 11,591 companies and found that most companies have fewer cash holdings, but companies with more cash generally perform better, and liquid cash provided material support for corporate contingencies or immediate conflicts [4,49].

Whether relational capital can directly improve corporate performance has not yet been concluded, but relational capital investment does have higher requirements on liquid cash and capital structure [16]. Therefore, it can be considered that the more current the assets are, the stronger the enterprise development, and the better the preventive effect of relational capital on sustainability risk will be. Based on this, we propose the following hypothesis:

**Hypothesis 3 (H3).** *Enterprise development has a positive moderating effect on relational capital and sustainability risk.*

2.2.4. Marketization, Relational Capital, and Sustainability Risk

According to the market competition hypothesis, the financial and strategic information of enterprises is transmitted transparently and rapidly in a competitive market, and the enterprise information reflected by the profit or price signal is true and reliable [50]. Drawing on Alchian's theory of economic change evolution, market competition is the most powerful force for obtaining economic efficiency, and marketization affects the achievement of strategic effects. In a sense, enterprises will consciously improve production and operation under the pressure of the external environment [51]. Combining the evolution of marketization and the characteristics of relational capital, Cullen et al. believe that trust and commitment are crucial elements of relational capital, and from an economic perspective, without the necessary trust, even the most promising relationships are likely to fail [8]. The construction of trust needs to weaken the asymmetry of information and improve the speed of information flow, and marketization can effectively promote the transmission of information. Therefore, marketization affects the formulation of enterprise strategy and the achievement of relational capital effects, and the impact of relational capital on sustainability risk may be heterogeneous with the marketization in the region where the firm is located [9,51]. Based on this, it can be inferred that market conditions will affect the role of relational capital, and in developed markets, relational capital will show a stronger mitigation effect on sustainability risk. Hence, we propose the following hypothesis:

**Hypothesis 4 (H4).** *Marketization has a positive moderating effect on relational capital and sustainability risk.*

*2.3. Theoretical Framework*

The overall research framework is divided into two parts. The first part aims to define the concept of relational capital and construct measurement indicators based on the actual situation of Chinese enterprises. Combining social capital theory and existing research results, we take the operating sequence as the dividing line. Specifically, we divide relational capital into (1) vertical relational capital, which includes customer relationships and supplier relationships; (2) horizontal relational capital, which includes concurrence relationships and other inter-enterprise relationships; and (3) parallel relational capital, which includes bank–enterprise relationships and external reputation. In this process, according to the mature index system of social capital, we assign quantitative values to relational capital variables and then use factor analysis to construct the index system.

The second part of the research framework analyzes the effects of relational capital. The paper uses a sample of Chinese non-state-owned manufacturing enterprises to explore the relationship between relational capital and sustainability risk. Based on agency theory, capital structure theory, and market competition hypothesis, we discuss the moderating effects of enterprise growth, enterprise development, and marketization on relational capital and sustainability risk. Our proposed model is shown in Figure 1.

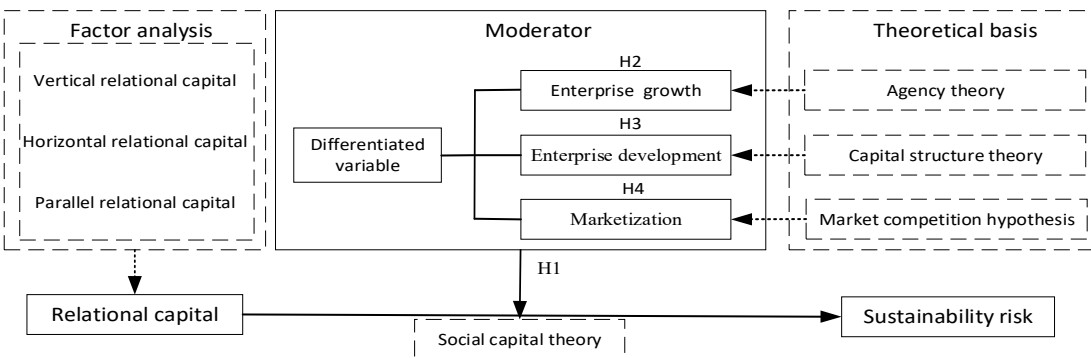

**Figure 1.** Theoretical framework.

The significance of our research is twofold. On the one hand, our work aims to improve the research on relational capital. Compared with Western markets, China has a profound clan ideology and kinship culture, and relational capital plays a more prominent role in Chinese enterprises. By examining this concept against a Confucian cultural and enterprise management, we can both analyze relational capital in a context where it plays a heightened role and strengthen our understanding of the features of this socialist market economy. On the other hand, our work enriches the literature on the effects of relational capital. Specifically, it improves our understanding of the relationship between relational capital and sustainability risk, enhances our understanding of the effects of relational capital on enterprise risk management, and makes up for some deficiencies in the current research base while simultaneously clarifying the operation of the socialist market economy, which has theoretical value and practical significance.

## 3. Material and Methods

### 3.1. Factor Analysis

Combined with existing research and management practices, we use the factor analysis method to construct a relational capital indicator to scientifically reflect the strength of corporate relational capital in the Chinese context.

3.1.1. Variable Measurement

Before the factor analysis method is used to construct an index system, all variables should be defined and assigned. Combined with the social capital measurement index and the characteristics of relational capital, after joint exploration and revision by an expert team, we determined the various elements of relational capital, as shown in Table 1. Using the direction of external communication as a distinguishing factor, we divided relational capital into vertical relational capital, horizontal relational capital, and parallel relational capital. What is more, vertical relational capital includes customer relationship and supplier relationship, horizontal relational capital includes concurrence relationship and other inter-enterprise relationship, and parallel relational capital includes bank–enterprise relationship and external reputations.

The enterprises used in our sample were selected from listed companies in the Shanghai and Shenzhen stock markets. To ensure the credibility and effectiveness of the research conclusions, we did not impose additional requirements based on enterprise characteristics. The relevant data were obtained from the WIND, CSMAR databases, and corporate annual reports. In terms of data collection, data were screened according to the basic standards: (1) firms in the financial and insurance industries were excluded; (2) listed companies with abnormal trading status were removed; (3) companies with missing or wrong data were removed. The time interval covered spans from 1 January 2015 to 31 December 2019, and valid data for a total of 6065 companies were included in the sample.

**Table 1.** Variable interpretation and assignment.

| Type | Interpretation | Symbol | Assignment |
|---|---|---|---|
| Vertical relational capital | Customer relationship | $X_1$ | Expressed as accounts receivable turnover |
| | Supplier relationship | $X_2$ | Expressed as accounts payable turnover |
| Horizontal relational capital | Concurrence relationship | $X_3$ | Expressed by the ratio of related party transaction amount to main business income |
| | Other inter-enterprise relationship | $X_4$ | Expressed by the proportion of senior executives working part-time in other companies |
| Parallel relational capital | Bank-enterprise relationship | $X_5$ | Expressed by the ratio of short-term Borrowings to current assets |
| | External reputation | $X_6$ | Expressed by the natural logarithm of intangible asset items |

Note: Related party transactions include commodities, assets, labor services, and agency transactions. Fund transactions, such as guarantees, counter-guarantees, and mortgages, are not included.

### 3.1.2. Feasibility Test

To determine whether a selected variable is eligible for factor analysis, we need to carry out a feasibility test. We used the SPSS software to conduct KMO tests and Bartlett sphericity tests on the original data [52]. The test results are shown in Table 2. Theoretically, factor analysis can be used when KMO > 0.5 and Bartlett significance < 0.005. In this study, KMO = 0.562, and Bartlett significance = 0.000, so it was feasible to use factor analysis.

**Table 2.** KMO test and Bartlett test results.

| KMO Sampling Suitability Quantity | | **0.562** |
|---|---|---|
| Bartlett sphericity test | Approximate $\chi^2$ | 871.550 |
| | df | 15 |
| | Significance | 0.000 |

### 3.1.3. Model Construction

In keeping with the principle of "eigenvalue > 1" in factor analysis, the principal components were extracted for the variable indicators. In our study, three common factors were selected to measure the strength of relational capital. After the common factors were extracted, the cumulative variance contribution rate exceeded 60%, which is effective. The specific results are shown in Table 3. Thus, the obtained principal components can explain the original variable indicators.

**Table 3.** Description of the total variance.

| | Initial Eigenvalues | | | Extract the Load Sum of Squares | | | Rotational Load Sum of Squares | | |
|---|---|---|---|---|---|---|---|---|---|
| | Total | Percent Variance | Cumulative | Total | Percent Variance | Cumulative | Total | Percent Variance | Cumulative |
| 1 | 1.432 | 23.870 | 23.870% | 1.293 | 21.552 | 21.552% | 1.293 | 21.548 | 21.548% |
| 2 | 1.153 | 19.216 | 43.086% | 1.037 | 17.278 | 38.830% | 1.033 | 17.212 | 38.760% |
| 3 | 0.972 | 17.204 | 60.290% | 1.009 | 16.817 | 55.647% | 1.013 | 16.888 | 55.647% |
| 4 | 0.870 | 14.498 | 73.788% | | | | | | |
| 5 | 0.838 | 13.959 | 87.747% | | | | | | |
| 6 | 0.735 | 12.253 | 100.000% | | | | | | |

Note: all values have three significant decimal places.

The data were then rotated by the maximum variance method to obtain a factor loading matrix, which reflected the relationship between the common factor and each variable. As shown in Table 3, the score of each factor was obtained by the regression method, and the ratio of the variance of each factor to the cumulative variance was used as the weight to calculate the comprehensive factor score. Finally, the relational capital (*Capital*) measurement index was obtained:

$$Capital = 0.160X_1 + 0.164X_2 + 0.154X_3 + 0.176X_4 + 0.188X_5 + 0.158X_6 \tag{1}$$

### 3.2. Empirical Analysis

Based on existing research, we empirically analyze the relationship between relational capital and sustainability risk and further explore the moderating effect of enterprise development, enterprise growth, and marketization.

#### 3.2.1. Data Collection

In our model of a socialist market economy with Chinese characteristics, the non-state-owned manufacturing industry shoulders most of the responsibility for China's industrial optimization and upgrading and faces greater pressure from market requirements and institutional norms. The effects of relational capital are more prominent, as is the impact of sustainability risk on operations. We believe the manufacturing industry best reflects the proposed characteristics of the socialist market economy, so we selected the Chinese non-state-owned manufacturing industry as the research context. Meanwhile, to ensure continuity and integrity, we selected companies for a sample that met the requirements used in the construction of the relational capital indicators but limited them to only those firms in non-state-owned and manufacturing industries. Our final sample included 1740 companies. In addition, to avoid the influence of extreme values on the hypothesis testing, we winsorized continuous variables at the 1% and 99% levels [53].

#### 3.2.2. Variable Definition

The explained variable in our study is sustainability risk, represented by Altman's Z-score index [54]. Compared with other indicators, the Z-score is more suitable for developing countries and has been repeatedly validated in this context [55]. It should be noted that the Z-score is an inverse index, so the smaller the value, the greater the possibility of corporate bankruptcy and the more serious the sustainability risk.

Our explanatory variable is relational capital, which is measured as Equation (1).

Our moderating variables are enterprise growth, enterprise development, and marketization. We use the Tobin $Q$ ratio to measure enterprise growth, the quick ratio to measure enterprise development, and the marketization index to measure marketization [36].

The control variables are identified by referring to the existing literature and included three types: (1) indicators that characterize the enterprise's financial status, including the leverage level and the cash ratio; (2) indicators that describe the enterprise's attributes, including enterprise size and age; and (3) the equity concentration, which reflects the corporate governance [9]. The formula and references for each variable are shown in Table 4.

**Table 4.** Variable definition and formula.

| Type | Variable | Symbol | Definition and Formula |
|---|---|---|---|
| Explained variable | Sustainability risk | Z-score | 1.2 × Net Working Capital/Total Assets + 1.4 × Retained Earnings/Total Assets + 3.3 × EBIT/Total Assets + 0.6 × Stock Market Value/Total Liabilities + 1.0 × Sales Revenue/Total Assets |
| Explanatory variable | Relational capital | Capital | Equation (1) |
| Moderating variables | Enterprise growth | Tobin $Q$ | (Market Value + Current Liabilities + Noncurrent Liabilities)/Total Assets |
| | Enterprise development | $QR$ | (Current Assets–Inventory)/Current Liabilities |
| | Marketization | $MI$ | Marketization Index Report by Provinces in China |
| | Equity concentration | Top 10 | The shareholding ratio of the top 10 shareholders |
| Control variables | Age | Age | Add 1 to the listing year, take the natural logarithm |
| | Size | Size | Natural logarithm of Total Assets at the end of the year |
| | Leverage level | Lev | Total Assets/Total Liabilities |
| | Cash ratio | CR | Cash Flow/Total Liabilities |

3.2.3. Hypothetical Model Construction

To verify H1, which describes the relationship between relational capital and sustainability risk, and construct the model, we calculated the Z-score (2):

$$Z - score = \alpha_0 + \alpha_1 Capital + \Sigma \alpha_i Controls_i + \varepsilon \tag{2}$$

To verify H2, which describes the relationship among relational capital, sustainability risk, and enterprise growth, we added the moderator variable Tobin *Q* to model (2) and constructed model (3):

$$Z - score = \alpha_0 + \alpha_1 Capital + \alpha_2 Tobin\ Q + \alpha_3 Capital \times Tobin\ Q + \Sigma \alpha_i Controls_i + \varepsilon \tag{3}$$

To verify H3, which focuses on the relationship between relational capital, sustainability risk, and enterprise development, we added the moderator variable *QR* to model (2) and constructed model (4):

$$Z - score = \alpha_0 + \alpha_1 Capital + \alpha_2 QR + \alpha_3 Capital \times QR + \Sigma \alpha_i Controls_i + \varepsilon \tag{4}$$

To verify H4, which addresses the relationship among relational capital, sustainability risk, and marketization, we added the moderator variable *MI* to model (2) and constructed model (5):

$$Z - score = \alpha_0 + \alpha_1 Capital + \alpha_2 MI + \alpha_3 Capital \times MI + \Sigma \alpha_i Controls_i + \varepsilon \tag{5}$$

In all models, *i* represents a non-state-owned manufacturing enterprise, controls represent a control variable group, and $\varepsilon$ is a random disturbance term.

## 4. Results

### 4.1. Descriptive Analysis

Descriptive analysis was conducted for all variables, and the results are shown in Table 5. According to the Z-score index explanation, when the Z-score < 1.81, the sustainability risk of the enterprise is considered dangerous, whereas when the Z-score > 2.99, the operation is considered stable and sustainable. The mean and median Z-scores of the sample companies were relatively high, indicating that their sustainability risk was generally stable, and they had a low risk of going bankrupt. However, both the standard deviation and the maximum difference were relatively large, reflecting the enterprises' uneven development and the need to control operational risks. The overall level of capital was low and the differences between companies were huge, which means that the enterprises place different emphases on relational capital and vary in their understanding of their external networks. A few companies had very high relational capital. Thus, how to properly handle the construction of relational capital needed further clarification.

**Table 5.** Descriptive statistical results.

|  | Mean | Median | Standard Deviation | Minimum | Maximum |
|---|---|---|---|---|---|
| Z-score | 5.025 | 4.009 | 3.405 | 0.891 | 20.486 |
| Capital | 6.624 | 4.619 | 10.089 | 3.262 | 114.759 |
| Tobin *Q* | 1.977 | 1.667 | 1.002 | 0.832 | 6.373 |
| *QR* | 1.446 | 1.196 | 0.894 | 0.230 | 5.168 |
| *MI* | 9.131 | 9.680 | 1.587 | 4.150 | 10.960 |
| Top 10 | 0.579 | 0.586 | 0.138 | 0.266 | 0.914 |
| Age | 2.840 | 2.890 | 0.296 | 2.079 | 3.497 |
| Size | 22.369 | 22.277 | 1.013 | 20.584 | 26.434 |
| Lev | 0.414 | 0.409 | 0.152 | 0.117 | 0.868 |
| CR | 0.462 | 0.313 | 0.473 | 0.033 | 2.670 |

The Tobin *Q* reflects the company's market value and growth status. The distribution of this value for the sample companies was slightly scattered, but the overall performance was upward. It is generally believed that a quick ratio of about one is more appropriate. If the quick ratio is too low, the company has weak short-term payment ability and faces a greater risk of being called upon for immediate debt repayment. The median and mean of the *QR* were slightly greater than one, while the standard deviation was about 0.9, indicating that these manufacturing enterprises had stable short-term solvency. However, there were significant differences in their marketization, reflecting their uneven regional distribution, and most non-state-owned manufacturing enterprises were in areas characterized by high marketization. The standard deviations of the control variables differed, which reflects differences between the enterprises in terms of their financial investment and business development, but also shows that the sample met the requirements for randomness.

### 4.2. Correlation Analysis

Table 6 shows the Pearson correlation coefficients for our data. The correlation coefficient between Capital and Z-score was significantly positive at the 1% level, indicating a positive correlation between these variables. Since Z-score is a reverse indicator when its value is large, the firm faces a more prominent sustainability risk; thus, having more relational capital may reduce the firm's sustainability risk—a finding that preliminarily confirms H1. Furthermore, the variance inflation factor (VIF) was less than two, and there was no multicollinearity problem among the variables.

**Table 6.** Correlation analysis.

| | Z-Score | Capital | Tobin Q | QR | MI | Top 10 | Age | Size | Lev | CR |
|---|---|---|---|---|---|---|---|---|---|---|
| Z-score | 1 | | | | | | | | | |
| Capital | 0.044 * | 1 | | | | | | | | |
| Tobin Q | 0.446 ** | −0.002 | 1 | | | | | | | |
| QR | 0.675 *** | −0.072 *** | 0.255 *** | 1 | | | | | | |
| MI | −0.003 | −0.069 ** | −0.040 * | 0.049 ** | 1 | | | | | |
| Top 10 | 0.079 ** | 0.063 ** | 0.013 | 0.058 ** | 0.121 *** | 1 | | | | |
| Age | −0.045 * | 0.036 | −0.101 *** | −0.029 | −0.021 | −0.062 ** | 1 | | | |
| Size | −0.386 *** | 0.144 *** | −0.372 *** | −0.288 *** | −0.093 *** | −0.042 * | 0.121 ** | 1 | | |
| Lev | −0.486 *** | 0.026 | −0.285 *** | −0.701 *** | −0.026 | −0.085 *** | 0.042 * | 0.465 *** | 1 | |
| CR | 0.470 *** | 0.017 | 0.265 *** | 0.775 *** | −0.015 | 0.047 * | −0.008 | −0.206 *** | −0.543 *** | 1 |

Note: ***, **, and * indicate significant at the 1%, 5%, and 10% levels, respectively, the same below.

### 4.3. Regression Analysis

The regression analysis results for the model are shown in Table 7. The first four columns (Models 1a–4a) show the interaction between relational capital and sustainability risk. Model 1a is the regression result including only Capital. The coefficient of Capital was 0.067, which was significantly positive at the 1% level, indicating that relational capital had a positive correlation with the Z-score and a significant negative correlation with sustainability risk. That is to say, the larger the relational capital, the higher the Z-score and the lower the sustainability risk. Thus, H1 was supported.

Models 2a, 3a, and 4a add Tobin *Q*, *QR*, *MI*, and their multiplication terms with Capital, respectively, to Model 1a to verify H2, H3, and H4. In Model 2a, the coefficient of Tobin *Q* was significantly positive at the 1% level, and the coefficient of the multiplication term with Capital was significantly positive at the 5% level, indicating that Tobin *Q* positively adjusts the relationship between Capital and Z-score. Thus, enterprise growth had a positive moderating effect on relational capital and sustainability risk, which supports H2. Similarly, in Model 3a, the coefficients of Capital, *QR*, and the intersection of capital and *QR* were all significantly positive at the 1% level, indicating that enterprise development had a positive moderating effect on relational capital and sustainability risk. This result supports H3. Finally, after adding the multiplication term of Capital and *MI* in Model 4a, its coefficient was still significantly positive at the 1% level, indicating that marketization positively regulated the relationship between relational capital and sustainability risk. Thus,

marketization can improve the value of relational capital and effectively resist operational risks. This finding supports H4.

**Table 7.** Regression model and lagging model results.

| | Model 1a | Model 2a | Model 3a | Model 4a | Model 1b | Model 2b | Model 3b | Model 4b |
|---|---|---|---|---|---|---|---|---|
| | Z-Score | Z-Score | Z-Score | Z-Score | Lagging Z-Score | Lagging Z-Score | Lagging Z-Score | Lagging Z-Score |
| Capital | 0.067 *** | 0.039 *** | 0.131 *** | 0.077 *** | 0.058 ** | 0.036 ** | 0.117 *** | 0.076 *** |
| Tobin $Q$ | | 0.596 *** | | | | 0.452 ** | | |
| Capital $\times$ Tobin $Q$ | | 0.022 ** | | | | 0.030 * | | |
| $QR$ | | | 0.336 *** | | | | 0.230 *** | |
| Capital $\times$ $QR$ | | | 0.087 *** | | | | 0.090 *** | |
| $MI$ | | | | −0.020 | | | | 0.013 * |
| Capital $\times$ $MI$ | | | | 0.040 *** | | | | 0.056 ** |
| Top 10 | 0.047 ** | 0.018 * | 0.049 ** | 0.048 ** | 0.096 *** | 0.073 *** | 0.096 *** | 0.096 *** |
| Age | −0.010 | 0.025 ** | −0.009 | −0.012 | 0.019 | 0.045 ** | 0.019 | 0.116 |
| Size | −0.112 *** | 0.075 *** | −0.122 *** | −0.115 *** | −0.117 *** | 0.025 | −0.124 *** | −0.116 *** |
| Lev | −0.480 *** | −0.456 *** | −0.346 *** | −0.477 *** | −0.375 *** | −0.357 ** | −0.285 *** | −0.372 *** |
| CR | 0.285 *** | 0.178 *** | 0.090 *** | 0.283 *** | 0.267 *** | 0.184 *** | −0131 *** | 0.266 *** |
| $F$ | 340.891 | 1086.346 | 293.710 | 257.230 | 192.064 | 286.820 | 155.639 | 145.777 |
| $R^2$ | 0.541 | 0.834 | 0.576 | 0.541 | 0.399 | 0.570 | 0.418 | 0.403 |
| Adjusted $R^2$ | 0.540 | 0.833 | 0.574 | 0.543 | 0.397 | 0.568 | 0.416 | 0.400 |
| $p$ | 0.000 | 0.000 | 0.000 | 0.000 | 0.000 | 0.000 | 0.000 | 0.000 |

Note: ***, **, and * indicate significant at the 1%, 5%, and 10% levels respectively.

Judging from the $R^2$, adjusted $R^2$, and F values, the proposed model had high goodness of fit with the variable data. Thus, the construction of the model was deemed reasonable and better able to explain the relationship between the variables in the hypothesis [56].

*4.4. Robustness Check*

To ensure the robustness of the regression results, we conducted three robustness tests. The specific operations and results are as follows.

First, we replaced the data for the explained variable. The construction of relational capital is a long-term process, which may have a lagging effect on sustainability risk. Therefore, we revised the sustainability risk data to include a lag period and then carried out the same regression analysis to eliminate possible endogeneity [57]. The regression results are shown in the last four columns of Table 7 (Models 1b–4b). After lagging the Z-score, the interaction terms between Capital and Tobin *Q*, *QR*, and *MI* were significantly positive at the 10%, 5%, and 1% levels, and relational capital was significantly negatively correlated with sustainability risk. Thus, enterprise growth, development, and marketization appeared to play positive roles in regulating relational capital and sustainability risk.

Second, we changed the form of the explanatory variable. Specifically, we subtracted the mean of relational capital from the variable data. If the result was greater than zero, indicating that the sample company's relational capital was higher than the industry average, we assigned it a value of one; otherwise, it was assigned a value of zero. The explanatory variables were changed from continuous variables to dummy variables and brought back into the regression test, yielding the results shown in the first four columns of Table 8 (Models 1c–4c). The results indicated that relational capital significantly reduced sustainability risk, while the multiplication of the moderator variables and explanatory variables maintained a positive correlation at the 1% level, indicating that the research conclusion was valid.

**Table 8.** Robustness test results.

|  | Model 1c | Model 2c | Model 3c | Model 4c | Model 1c | Model 2d | Model 3d | Model 4d |
|---|---|---|---|---|---|---|---|---|
|  | Z-Score | Z-Score | Z-Score | Z-Score | Z-Score | Z-Score | Z-Score | Z-Score |
| Capital | 0.067 *** | 0.039 ** | 0.126 *** | 0.077 *** | 0.063 *** | 0.039 ** | 0.087 *** | 0.075 *** |
| Tobin $Q$ |  | 0.596 *** |  |  |  | 0.596 *** |  |  |
| Capital $\times$ Tobin $Q$ |  | 0.022 ** |  |  |  | 0.022 ** |  |  |
| QR |  |  | 0.330 *** |  |  |  | 0.336 *** |  |
| Capital $\times$ QR |  |  | 0.088 *** |  |  |  | 0.131 *** |  |
| MI |  |  |  | −0.020 |  |  |  | −0.021 |
| Capital $\times$ MI |  |  |  | 0.040 ** |  |  |  | 0.043 ** |
| Top 10 (Top 1) | 0.047 ** | 0.026 ** | 0.015 | 0.048 ** | 0.016 | 0.026 ** | 0.049 ** | 0.020 |
| Age | −0.010 | 0.027 ** | −0.007 | −0.012 | −0.009 | 0.027 ** | −0.009 | −0.011 |
| Size | −0.112 *** | 0.076 *** | −0.119 *** | −0.115 *** | −0.111 *** | 0.075 *** | −0.122 *** | −0.113 *** |
| Lev | −0.480 *** | −0.454 *** | −0.349 *** | −0.477 *** | −0.480 *** | −0.456 *** | −0.346 *** | −0.477 *** |
| CR | 0.285 *** | 0.178 *** | 0.095 *** | 0.283 *** | 0.286 *** | 0.178 *** | 0.090 *** | 0.284 *** |
| $F$ | 340.891 | 1087.562 | 0.572 | 0.541 | 337.490 | 1086.346 | 0.576 | 0.543 |
| $R^2$ | 0.541 | 0.834 | 0.570 | 0.539 | 0.539 | 0.834 | 0.574 | 0.541 |
| Adjusted $R^2$ | 0.540 | 0.833 | 289.658 | 254.917 | 0.537 | 0.833 | 293.710 | 257.230 |
| $p$ | 0.000 | 0.000 | 0.000 | 0.000 | 0.000 | 0.000 | 0.000 | 0.000 |

Note: *** and ** indicate significant at the 1% and 5% levels respectively.

Finally, we changed the control variables. We replaced the shareholding ratio of the top 10 shareholders as a control variable with the shareholding ratio of the top one shareholder and then conducted regression analysis. The results are shown in the last four columns of Table 8 (Models 1d–4d). The coefficient of Capital was significantly positive, and the multiplication terms of Tobin *Q*, *QR*, *MI*, and Capital were all significantly positively correlated with the Z-score, which also verifies that the results obtained were robust.

## 5. Discussion

### 5.1. Empirical Findings

Under the guidance of social capital theory, we redefined the definition of relational capital and used factor analysis to construct an index system of relational capital in the context of China. On this basis, we empirically tested the interaction between relational capital and sustainability risk. The research suggested that relational capital has an effective defensive effect on sustainability risk. Moreover, enterprise growth, enterprise development, and marketization play a moderating role between relational capital and sustainability risk.

Of note, the conclusions of this study show that relational capital can significantly mitigate the sustainability risk by the regression analysis (Hypothesis 1). The reason is that the creation of relationships itself is a productive activity, and enterprises realize the transmission of information and resource sharing by building relationship networks [6]. Relational capital is a real or potential resource generated and existing in the enterprise relationship network, which promotes information sharing and improves organizational stability, to slow down the sustainability risk of enterprises and ensure the sustainability of the operation [10]. Indeed, competitive market strategies such as product innovation and business expansion can effectively resist sustainability risks and are the main factors to ensure the stability of enterprise operation. However, the factors affecting the development of enterprises are diverse, and relational capital still plays a role that cannot be ignored. Even in some areas with imperfect social credit, it directly determines the survival of enterprises [37].

This study demonstrates that enterprise growth can strengthen the resistance of relational capital to sustainability risk (Hypothesis 2). This is because enterprise growth usually reflects the enterprise value and management ability. When an enterprise is in a high-level growth state, the quality of relational capital is more prominent [43]. At this point, the enterprise has flexibly mastered the ability to use relational capital and can skillfully convert relational capital into resources required for operation and then use it efficiently. Besides, regression research shows that enterprise development can also improve

the ability of relational capital to prevent sustainability risk (Hypothesis 3). Enterprise development is the expression of the internal financial and capital structure, and the quick ratio of assets directly affects the investment intensity of the relational capital and the reaction speed of strategic change. Once the transfer and realization of relational capital are required, the enterprise can make emergency changes immediately, so it can more effectively resist the sustainability risk [48]. In addition, empirical results confirm that market conditions also affect the effectiveness of relational capital in resisting sustainability risk (Hypothesis 4). Classical strategic management points out that the effect of strategic decision making is profoundly affected by the external environment [58]. There is a regularity and fairness mechanism in high marketization, so the competition of enterprises depends more on strategic decision making and resource allocation ability. Therefore, relational capital exists reasonably and transparently, and enterprises can decide the arrangement of relational capital independently, so the effectiveness of resisting sustainability risk can be fully demonstrated.

*5.2. Research Contribution*

In the study, our main work is to construct relational capital indicators and empirically test the relationship between relational capital and sustainability risk in the Chinese context. The research contributions mainly include the following two points.

First, influenced by Confucian culture, Chinese companies attach great importance to relational capital while following the market system. In turn, relational capital has a more profound effect in this environment, which is a unique feature that distinguishes China from Western markets. However, there is little research on relational capital in the context of China, so it is impossible to correctly grasp the significance of relational capital in Chinese enterprises. With the help of social capital theory, we redefine the concept of relational capital, emphasizing the equal status of both parties in the relationship and excluding government–enterprise exchanges, which is different from the past. Relational capital is the resources generated by a company's interactions with external entities of equal status, which are aimed at enhancing competitive advantage. Then, we used the factor analysis method to construct a relational capital index including six types of elements, thereby extending prior research that provides a new perspective for scientifically evaluating relational capital. The definition and measurement of relational capital strengthen the cognition of relational capital and the understanding of the socialist market economy, so it fills the research gap to a certain extent, which is a major innovation of this study.

Second, we enrich research on the effects of relational capital. Notably, most of the existing literature on relational capital has focused on strategic alliances and supply chains, and the few effect studies are also limited to performance or innovation, ignoring its relationship with sustainability risk, which is an important reflection of long-term sustainable development. The value of relational capital remains unclear, hindering the constructive development of relational capital theory. We used Altman's Z-score index to measure sustainability risk and explored the connection between relational capital and sustainability risk—a thematic supplement to the existing literature that has theoretical significance. The regression test results theoretically confirm the resistance of relational capital to sustainability risk and further confirm that enterprise growth, enterprise development, and marketization have a significant positive moderating effect on it. Our conclusions affirm the significance of relational capital in enterprise management from an empirical perspective, change the negative view of this relationship, and supplement the existing theories, thereby guiding enterprises to strengthen their cultivation of relational capital.

## 6. Conclusions

*6.1. Research Summary*

We summarized the research on relational capital and draw out the theoretical gap and practical necessity of the in-depth research on it in Chinese enterprises. Then, based on social capital theory and specific management practices, we redefined the concept

of relational capital in the Chinese context, highlighting the equality and interactions in this realm and classifying government–enterprise interactions as a type of political strategy. Taking advantage of the mature measurement system for social capital, we constructed a measurement index of relational capital through factor analysis from the six perspectives of the customer relationship, supplier relationship, concurrence relationship, other inter-enterprise relationships, bank–enterprise relationship, and external reputation. This approach provides a new method for the scientific measurement of relational capital.

In addition, to further understand the role of relational capital in the socialist market economy, we empirically tested the relationship between relational capital and sustainability risk in a sample consisting of Chinese non-state-owned manufacturing enterprises. The results showed that relational capital can effectively reduce the bankruptcy risk faced by such businesses. Moreover, enterprise growth, enterprise development, and marketization positively regulate the relationship between relational capital and sustainability risk. That is to say, the higher the enterprise value, the stronger the short-term solvency, or the more developed the market, the more effective the role of relational capital will be and the better the firm will be able to mitigate sustainability risk. Our research confirms the positive impact of relational capital from a theoretical perspective and suggests that companies can mitigate their operational risks by building relational capital.

### 6.2. Management Implications

Our results led to several management suggestions from the perspectives of enterprises, governments, and managers.

First, while strengthening market competitiveness, enterprises also need to improve their understanding and application of relational capital and non-market strategies. Relationship networks and interpersonal communications have emerged as external factors that cannot be ignored in the development of enterprises, especially for those operating in areas with strong kinship culture or immature market development. The acquisition and maintenance of an enterprise's sustainable competitive advantage depend not only on market strategies but also on the support and recognition of external informal systems. Only by effectively integrating market strategies and non-market strategies, and rationally applying their relational capital, can companies acquire strategic resources and improve their ability to resist sustainability risk.

Second, as a market regulator, the government should deepen regional marketization and guide enterprises to build relational capital rationally. Public management theory points out that the government has the responsibility to ensure social affairs run smoothly and to promote social development by integrating social resources and using political or economic means. Marketization affects settlement choices and strategic effects, so the government must comprehensively improve regional marketization to build an open market and a transparent institutional environment. In addition, the government should seek to enhance management efficiency and strive to attract enterprises to settle in certain areas through tax incentives and financial subsidies, to ensure the positive effect of relational capital while also blocking enterprise rent-seeking and institutional corruption.

Third, for enterprise managers, relational capital can reduce sustainability risk and help companies gain competitive advantages in an informal environment. To realize these benefits, managers must strengthen cognition of relational capital and improve sustainable competitive advantage by integrating market and non-market strategies. What is more, in the construction of relational capital, managers should focus on strengthening the growth and development of their enterprises, so that relational capital can achieve its maximal intentions. It should be noted that managers must reasonably grasp the concepts underlying relational capital and cannot blindly rely on relational capital as the sole means to seek development resources. From a factual standpoint, the market is still the main field of corporate competition, and relational capital can play an effective role only when it is combined with the appropriate market strategies.

*6.3. Limitations and Future Research*

Based on the social capital theory and practices employed in China's socialist market economy, our research redefined the concept of relational capital and used factor analysis to construct relational capital measurement indicators. In addition, we empirically confirmed that relational capital can effectively mitigate sustainability risk and affirmed the value of relational capital. Though our work represents a valuable supplement to the existing theoretical research, some gaps persist in this literature.

First, when constructing the relational capital measurement index, we included external relationships with greater influence as part of the measurement. For the sake of typicality and pertinence, we focused on six major types of corporate relationships. However, corporate relationships are complicated, and many small relationships affect a firm's operations. Our paper does not discuss other, perhaps more subtle relationships. Second, when investigating the relationship between relational capital and sustainability risk, we considered only the impact of the enterprise itself on business risk but did not consider the impact of the strategic responses of other entities on sustainability risk. For example, in the customer relationship, whether consumers respond to relational capital will affect the realization of the proposed sustainability risk effect.

In future research, scholars might therefore enrich our understanding from three aspects. First, they can continuously deepen and refine the external relationship network of the enterprise and incorporate more subtle relationships into relational capital to improve its adaptability. Second, they can strengthen the influence of other entities' strategic responses on the effect of relational capital and build a strategic feedback chain with relational capital as the center to further enrich the research content. Third, they can enrich data testing, expand the application scope of relational capital concepts and theories, and improve the universality of the research conclusions.

**Author Contributions:** Conceptualization, D.Z. and H.W.; methodology, H.W.; software, H.W.; validation, D.Z., H.W. and W.W.; formal analysis, H.W.; investigation, H.W.; resources, D.Z.; data curation, H.W.; writing—original draft preparation, H.W.; writing—review and editing, W.W.; visualization, H.W.; supervision, W.W.; project administration, W.W.; funding acquisition, D.Z.; All authors have read and agreed to the published version of the manuscript.

**Funding:** This research was funded by the National Social Science Fund Project of China (17BJY033).

**Institutional Review Board Statement:** Not applicable.

**Informed Consent Statement:** Not applicable.

**Data Availability Statement:** The data presented in this study are available on request from the corresponding author.

**Conflicts of Interest:** The authors declare no conflict of interest.

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
