# Peer review of "The Influence of Relational Capital on the Sustainability Risk: Findings from Chinese Non-State-Owned Manufacturing Enterprises"

_sustainability, doi:10.3390/su14116904_

Round 1
Reviewer 1 Report
The manuscript of this article has problems that are very interesting to study, it's just that there are several things that become input for the manuscript, including:
1. When writing the abstract, please include interesting information related to the content of the article manuscript.
2. Introductory section, requires a strong GAP Research.
3. Method section, it is necessary to confirm the approach used
4. The Results and Findings section contains comprehensive data related to the research that has been carried out.
5. Discussion, This section must be made separately and discussed in depth, because in the manuscript there are still very few interpretations of the data findings.
6. Conclusion, made separately
7. Reference management, classified as not good, please fix it
For more details, please see the manuscript of the article which already contains the parts that need to be improved.

Author Response
Response to Reviewer 1 Comments
Thank you very much for your review and valuable suggestions, which are worth thinking deeply about. In response to your suggestion, I have carefully revised and supplemented the paper. In addition, we professionally polish articles to better match language style and writing norms. The following is my point-by-point explanation.
Point 1: When writing the abstract, please include interesting information related to the content of the article manuscript.
Response 1: We have revised the abstract and highlighted the conclusions and contributions. The description of the research conclusions is as follows:
The empirical results show that relational capital can effectively reduce sustainability risk and ensure sustainable operation. In addition, enterprise growth, enterprise development, and marketization can strengthen the role of relational capital and positively regulate the relationship between relational capital and sustainability risk. This paper innovatively constructs the concept and index system of relational capital in the Chinese context, which is the perfection of relational capital theory. At the same time, it verifies the impact of relational capital on business sustainability, revises the correct cognition of relational capital, and supplements the deficiencies of the extant social socialist market economy research. Supported by both theoretical research and empirical conclusions, corresponding management suggestions are put forward for enterprises, governments, and managers, to scientifically guide management practice and provide new ideas for future Chinese-style economic research.
Point 2: Introductory section, requires strong GAP Research.
Response 2: We have made major revisions to the “1. Introduction”, mainly removing irrelevant content and adding theoretical research review and existing research deficiencies to highlight the gaps in relational capital research. Existing research on increased capital relations, such as:
Roughly, relational capital is the sum of actual or potential resources embedded in, derived from, and emerging in external relational networks. Relational capital brings assets through the creation and utilization of relationships, which make relationships become resources to achieve individual and collective goals. Most of the research literature on relational capital is placed in specific organizational relationships, such as supply chains and corporate alliances, and less in a national environment. As a developing new theory, the future research of relational capital still needs to be improved continuously.
Capello & Faggian pointed out that relational capital is influenced by regional culture and may have different forms and manifestations in different regions, and even organizational structure and strategic orientation will have a significant impact on relational capital. To explore the theory and application of relational capital in the Chinese context, it is necessary to start from China's specific economic and cultural background and form its theoretical framework of relational capital.
……
Point 3: Method section, it is necessary to confirm the approach used
Response 3: According to the expert's opinion, we readjusted the chapter arrangement of "3. Material and Methods", confirm the research methods, and add explanations to them. The adjusted chapter structure is “3. Material and Methods”; “3.1. Factor Analysis”; “3.2. Empirical Analysis”.
Point 4: The Results and Findings section contains comprehensive data related to the research that has been carried out.
Response 4: We have revised the title of the paper, and incorporated data analysis and empirical testing into the result. The structure of the article is revised to 4. Results, including 4.1. Descriptive Analysis; 4.2. Correlation Analysis; 4.3. Regression Analysis; 4.4. Robustness Check. The content of each subsection has also been revised.
Point 5: Discussion, this section must be made separately and discussed in-depth, because in the manuscript there are still very few interpretations of the data findings.
Response 5: We have separated and reinforced the discussion, which includes “5.1 Empirical Findings” and “5.2 Research Contributions”. Among them, the empirical results are the interpretation of the empirical conclusions to strengthen research analysis and highlight the role of relational capital; the research contribution is the description of the contribution and innovation, and its purpose is to highlight the significance of the research.
Point 6: Conclusion, made separately
Response 6: Following the proposed revisions, we also have presented our conclusions separately, including the research summary, management implications, and limitations, and then delete and revise the original content to make the article language more concise.
Point 7: Reference management, classified as not good, please fix it
Response 7: We have reformatted the citations for consistency. Additionally, we removed duplicate citations and added new ones based on the paper. There are 54 references in total after revision.
Thanks again for your review. I hope this revision is satisfactory!

Reviewer 2 Report
- The title: The Influence of Relational Capital on the Sustainable Risk: Findings from Chinese Non-State-Owned Manufacturing Enterprises
The paper analyzes the role of relational capital in the context of Chinese companies, but does not relates it to sustainability risk, as the title promises. The term “sustainable risk” appears 54 times in the paper, but we cannot find any proper definition of the term (should be rather “sustainability risk”). The paper also lacks proper literature revive on sustainability and sustainability risk.
The paper states:” Sustainable risk refers to the risk of business closure or bankruptcy faced by accounting entities when they operate in accordance with established strategic guidelines and management forms” – but this is rather a bankruptcy risk.
In the literature, sustainability is typically defined as: encouraging businesses to frame decisions in terms of financial, environmental (including climate, biodiversity), social and human effects, ensuring resilience and long-term value creation – the paper does not refer directly to any those issues.
Therefore, I suggest to change the title to something like: The Role of Relational Capital for Chinese Privately-Owned Manufacturing Enterprises
- Paper’s structure and theoretical foundations
The first part of the paper is poetry, rather than scientific research. The initial part talking on “Chinese unique culture, based on Confucian clan ideology” (I would prefer to refer to it as “Confucian philosophy”) is well written, but it is not supported by references or data and has nothing to do with the empirical part of the paper.
The literature on sustainability risk is extensive, and completely ignored by this paper. The author(s) states that “modern management theories are typically born in the mature Western markets, and it is inappropriate and one-sided to apply the theoretical conclusions from those markets to the development of Chinese enterprises without considering the cultural background” – which is a nice excuse to completely ignore a proper literature review.
3. Empirical research
The empirical research is interesting and the attempt to construct a relational capital index is a main value added of this paper. The hypotheses should be rephrased and concentrated on the relational capital and its role.
Author Response
Response to Reviewer 2 Comments
Thank you very much for your review and valuable suggestions, which are worth thinking deeply about. In response to your suggestion, I have carefully revised and supplemented the paper. In addition, we professionally polish articles to better match language style and writing norms. The following is my point-by-point explanation.
Point 1: The title: The Influence of Relational Capital on the Sustainable Risk: Findings from Chinese Non-State-Owned Manufacturing Enterprises
The paper analyzes the role of relational capital in the context of Chinese companies but does not relates it to sustainability risk, as the title promises. The term “sustainable risk” appears 54 times in the paper, but we cannot find any proper definition of the term (should be rather “sustainability risk”). The paper also lacks proper literature revive on sustainability and sustainability risk.
The paper states:” Sustainable risk refers to the risk of business closure or bankruptcy faced by accounting entities when they operate in accordance with established strategic guidelines and management forms” – but this is rather a bankruptcy risk.
In the literature, sustainability is typically defined as encouraging businesses to frame decisions in terms of financial, environmental (including climate, biodiversity), social and human effects, ensuring resilience, and long-term value creation – the paper does not refer directly to any of those issues.
Therefore, I suggest changing the title to something like The Role of Relational Capital for Chinese Privately-Owned Manufacturing Enterprises
Response 1: First, in “2.1.2. Sustainability Risk and Relational Capital”, we added the definition and interpretation of sustainability, risk, and sustainability risk. The supplementary content is as follows:
Sustainability can be considered as the degree to which present decisions of o-organizations impact the future situation of the natural environment, societies, and business viability. From a corporate perspective, sustainability is how a firm grows and develops its recognition of environmental, social, and economic issues, and encourages businesses to frame decisions in financial, environmental (including climate, biodiversity), social and human effects, to ensure resilience and long-term value creation. In Dyllick and Hockerts’ terms, this means corporations must meet their current objectives without diminishing their ability to meet the needs and demands of future stakeholders. Risk refers to the possibility of certain unfavorable elements arising from organizational structure and commitments, resulting in uncertainty and harm. Due to the multifaceted of sustainability, sustainability risk is viewed as a comprehensive consideration of environmental risks, economic risks, and social risks. Among them, Anderson defines sustainability risk as potential financial or operational crises in corporate growth arising from liability litigation, consumer boycotts, shareholder action, and international pressure.
Second, according to expert advice, though reading and comparing the literature, we confirmed that “sustainability risk” is more appropriate in this study, so we changed the original “sustainable risk” to “sustainability risk”. Besides, we explained the reasons for using “sustainability risk” in the study. ”Corporate sustainability risk roughly involves the environment, economy, and society, and is itself a complex concept. In this study, we focus on the analysis of sustainability risk in corporate operations and adopt Altman's research to define sustainability risk as the risk of bankruptcy and unsustainable operations that may occur when companies operate following established strategic guidelines and management forms. It is mostly used to judge the stability of the enterprise’s long-term operations.”
Third, we did not change the title of the article after taking into account your opinions and the research content, hoping to gain your understanding and approval. There are two main reasons: on the one hand, there are subtle differences between non-state-owned enterprises and private enterprises. In the composition of Chinese enterprises, non-state-owned enterprises include not only private enterprises, but also foreign-funded enterprises (although the proportion of foreign-funded enterprises is very small). For the sake of rigor, we maintain the expression of non-state-owned manufacturing. On the other hand, the main content of this study is to analyze the relationship between relational capital and sustainability risk. Sustainability risk is only a small part of the role of relational capital, and the role of relational capital is a broader analysis. Given the size and depth of the research, it may not be fit for a more ambitious title, although we also hope to fully analyze the role of relational capital.
Point 2: Paper’s structure and theoretical foundations
The first part of the paper is poetry, rather than scientific research. The initial part talking about “Chinese unique culture, based on Confucian clan ideology” (I would prefer to refer to it as “Confucian philosophy”) is well written, but it is not supported by references or data and has nothing to do with the empirical part of the paper.
The literature on sustainability risk is extensive and completely ignored by this paper. The author(s) states that “modern management theories are typically born in the mature Western markets, and it is inappropriate and one-sided to apply the theoretical conclusions from those markets to the development of Chinese enterprises without considering the cultural background” – which is a nice excuse to completely ignore a proper literature review.
Response 2: The introduction is indeed too poetic and lacks the theoretical of scientific research. Therefore, we have revised the relevant content to strengthen the analysis of existing research and highlight theoretical gaps. In addition, we corrected "Chinese unique culture, based on Confucian clan ideology" to "Confucian philosophy".
Capello & Faggian pointed out that relational capital is influenced by regional culture and may have different forms and manifestations in different regions, and even organizational structure and strategic orientation will have a significant impact on relational capital. To explore the theory and application of relational capital in the Chinese context, it is necessary to start from China's specific economic and cultural background and form its theoretical framework of relational capital.
Besides, we have added dialogue with existing research in both the “2.1. Theoretical Basis” and “Discussion” to improve the theoretical richness of this paper.
Research by Capello and Faggian shows that relational capital does not play a consistent role in different regions or different firm characteristics and the effect of relational resources is closely related to the state of the enterprise; Cullen et al. believe that trust and commitment are crucial elements of relational capital, and from an economic perspective, without the necessary trust and commitment, even the most promising relationships are likely to fail.
……
Point 3: Empirical research
The empirical research is interesting and the attempt to construct a relational capital index is the main value added of this paper. The hypotheses should be rephrased and concentrated on the relational capital and its role.
Response 3: We revise "2.2. Hypotheses", mainly by adding existing research conclusions to support the hypothesis and reinterpreting the theoretical framework to highlight the significance of the research. In addition, we have added a new section "5.1. Empirical Findings" to explain the findings separately, which highlights relational capital and its role.
5.1. Empirical Findings
Under the guidance of the social capital theory, we use factor analysis to construct an index system of relational capital in the context of China. On this basis, we empirically test the interaction between relational capital and sustainability risk. The research suggests that relational capital has an effective defensive effect on sustainability risk. Moreover, enterprise growth, enterprise development, and marketization play a moderating role between relational capital and sustainability risk.
Of note, the conclusions of this paper show that relational capital can significantly mitigate the sustainability risk by the regression analysis (hypothesis 1). The reason is that the creation of relationships itself is a productive activity because enterprises realize the transmission of information and the sharing of resources by building external relationship networks. Relationship capital is a real or potential resource generated and existing in the enterprise relationship network, which promotes information sharing and improves organizational stability, to slow down the sustainability risk of enterprises and ensure the sustainability of the operation. Indeed, competitive market strategies such as product innovation and business expansion can effectively resist sustainability risks and are the main factors to ensure the stability of enterprise operation. However, the factors affecting the development of enterprises are diverse, and relational capital still plays a role that cannot be ignored. Even in some areas with im-perfect social credit, it directly determines the survival of enterprises.
This study demonstrates that enterprise growth can strengthen the resistance of relational capital to sustainability risk (hypothesis 2). This is because enterprise growth usually reflects the enterprise value and management ability. When an enterprise is in a high-level growth state, the quality of relational capital is more prominent. At this point, the enterprise has flexibly mastered the ability to use relational capital, and can skillfully convert relational capital into resources required for operation and then use it efficiently. Besides, regression research shows that enterprise development can also improve the ability of relational capital to prevent sustainability risk (hypothesis 3). Enterprise development is the expression of the internal financial and capital structure, and the quick ratio of assets directly affects the investment intensity of the relational capital and the reaction speed of strategic change. Once the transfer and realization of relational capital are required, the enterprise can make emergency changes immediately, so it can more effectively resist the sustainability risk. In addition, empirical results confirm that market conditions also affect the effectiveness of relational capital in resisting sustainability risk (hypothesis 4). Classical strategic management points out that the effect of strategic decision-making is profoundly affected by the external environment. There is a regularity and fairness mechanism in high marketization, so the competition of enterprises depends more on strategic decision-making and resource allocation ability. Therefore, relational capital exists reasonably and transparently, and enterprises can decide the arrangement of relational capital independently, so the effectiveness of resisting risks can be fully demonstrated.
Thanks again for your review. I hope this revision is satisfactory!

Reviewer 3 Report
Nice work.
Author Response
Response to Reviewer 3 Comments
Thank you very much for your review, I am very happy to get your affirmation!
According to the comments of the other two experts, I have made some modifications to the article. The revisions mainly focus on three aspects. First, in "1. Introduction" and "2. Theoretical Basis and Hypotheses", we have added dialogue with existing research to enrich the theoretical value of the article (Expert 1's suggestion). Second, to improve the completeness of the article, we have supplemented the content in "5.1. Empirical Findings" and "5.2. Research Contribution" (Expert 2's suggestion). In the main part of the article, we have not made any changes to factor analysis and empirical analysis. Finally, in order to be more in line with the English reading habits and language logic, we have polished the article, so some words and sentences have been changed.
Thank you again!

Round 2
Reviewer 1 Report
This article deserves to be published, thanks to the very significant improvements made by the author. Prior to publication, there were several inputs regarding the use of reference styles, especially in the management of references and sources used (I have attached the complete input directly in the manuscript).

Author Response
Response to Reviewer 1 Comment
I am pleased to get your suggestions. Under your guidance, the article has indeed been revised more completely and logically. I am also sincerely grateful for your review during this time. Your recognition has inspired me and given me full confidence in the next scientific research.
Point: This article deserves to be published, thanks to the very significant improvements made by the author. Prior to publication, there were several inputs regarding the use of reference styles, especially in the management of references and sources used (I have attached the complete input directly in the manuscript).
A little I want to add and be input in the management of references and reading sources, in the use of reference styles, please note between the citation addresses:
> If using the IEEE style: the citation address [1], [2-4], and so on, while in the bibliography the order of use and number [1] is followed by the reference data and so on.
Example:
Just an example quote [1]
In the bibliography:
[1] D. Movilla-quesada, J. Rojas-mora, and A. C. Raposeiras, “Statistical Studies Based on the Kriging Method and Geographical Mapping on Rigid Pavement Defects in Southern Chile,” Sustain., vol. 14, no. 1, p. 1–14, 2022, doi:10.3390/su14010585.
> If using Vancover style, then in the quotation address (1), (2-4) and so on, while in the bibliography the order of use and number is 1. followed by reference data, and so on
Example:
Just an example quote (1)
In the bibliography:
- Movilla-quesada D, Rojasmora J, Raposeiras AC. Statistical Study Based on Kriging Method and Geographical Mapping on Rigid Pavement Defects in Southern Chile. maintain. 2022;14(1):1–14.
>> In this script there is a mixture of styles, in quotes using IEEE namely [1], [2] and so on. while in the bibliography using vancover style 1. 2. etc.
>>> Please choose which one to use, whether IEEE style or Vancover Style. (Same usage in citations as well as in bibliographies - Note the very different use of parentheses, placement of years in bibliographies, etc.)
Response: Thanks for the detailed advice and explanation. I re-checked the reference format on the journal's official website and revised the reference format of the paper according to the references of the latest article.
Reference format required by the official website:
Reference format for the latest article:
According to the above requirements, the reference format of the literature may be as follows:
- Movilla-quesada, D.; Rojasmora, J.; Raposeiras, A. C. Statistical study based on kriging method and geographical mapping on rigid pavement defects in Southern Chile. Sustainability. 2022, 14, 1–14.
I have reformatted all references and I guaranteed that the current format is in line with the journal requirements.
In addition, I modified the language of the article and deleted redundant or unclear sentences to enhance the logic of the research.
I hope my modification is satisfactory. Thank you again for your review and guidance!

Reviewer 2 Report
This version is much improved and although some statements / assumptions are still controversial, they are not incorrect.

Author Response
Response to Reviewer 2 Comment
I am pleased to get your suggestions. Under your guidance, the article has indeed been revised more completely and logically. I am also sincerely grateful for your review during this time. Your recognition has inspired me and given me full confidence in the next scientific research.
Point: This version is much improved and although some statements/ assumptions are still controversial, they are not incorrect. Is the content succinctly described and contextualized with respect to the previous and present theoretical background and empirical research (if applicable) on the topic?
Response: (1) I have revised the hypothesis section to strengthen the connection between the theory and the hypothesis. The content added in the hypothesis section mainly includes two aspects: one is the important research conclusions or core connotations under the theoretical basis; the other is the latest or vital conclusions related to the theory and research theme. Each hypothesis adds the following:
2.2.1 Relational Capital and Sustainability Risk
“Especially in an environment where China emphasizes informal institutional arrangements, relational capital can often bring important competitive advantages to enterprises, such as the latest policy developments or various preferential resource conditions, which can effectively resist business risks [37]. Moreover, Huang et al. also proved that relational capital can help determine market positioning and maintain organizational stability [38] ……Additionally, the research of Roehrich et al. also pointed out that business conditions will directly affect the stability and sustainability of enterprises [35]”
2.2.2 Enterprise Growth, Relational Capital, and Sustainability risk
“Williamson integrated agency theory and corporate finance theory and pointed out that the choice of corporate governance is not only affected by transaction costs but also depends on the governance structure and financial reflection [41]. The enterprise can improve its growth ability by optimizing the allocation of resources to achieve sustainable development. Research by Capello and Faggian showed that relational capital does not play a consistent role in different firm characteristics and the effect of relational resources is closely related to financial performance [9,42]. ……Fazzari et al. took American enterprises as samples to test the relationship between corporate growth and management performance and confirms that enterprises with excellent growth do indeed have better corporate investment performance [43]. Similarly, Hayashi et al. t conducted experiments with Japanese companies as research objects and came to the same conclusion [44].”
2.2.3. Enterprise Development, Relational Capital, and Sustainability Risk
“Agiomirgianakis et al. conducted regression analysis on Greek manufacturing data, and the results showed that the efficiency of capital structure and capital management affected the company's return rate [48]. In capital structure management, asset liquidity affects the enterprise’s ability to pay its debts and respond to strategic changes, which together guarantee its sustainable operation. Dittmar et al. studied the data of 11,591 companies and found that most companies have fewer cash holdings, but companies with more cash generally perform better, and liquid cash provided material support for corporate contingencies or immediate conflicts [4,49].”
2.2.4. Marketization, Relational Capital, and Sustainability Risk
“Drawing on Alchian 's theory of economic change evolution, market competition is the most powerful force for obtaining economic efficiency, and marketization affects the achievement of strategic effects. In a sense, enterprises will consciously improve production and operation under the pressure of the external environment [51]. Combining the evolution of marketization and the characteristics of relational capital, Cullen et al. believe that trust and commitment are crucial elements of relational capital, and from an economic perspective, without the necessary trust, even the most promising relationships are likely to fail [8]. The construction of trust needs to weaken the asymmetry of information and improve the speed of information flow, and marketization can effectively promote the transmission of information.”
(2) In addition, I modified the language of the article and deleted redundant or unclear sentences to enhance the logic of the research. This part of the revision mainly focuses on "2. Theoretical Basis and Hypotheses" and "3. Material and Methods".
(3) Finally, to improve smoothness and relevance, I adjusted and condensed the order of some of the contents of "2.1. Theoretical Basis”. Specifically, I put the overall analysis on the role of relational capital in "2.1.1. Social Capital Theory and Relational Capital", and "2.1.2. Sustainability Risk and Relational Capital" focuses on the analysis of sustainability risk and the relationship between relational capital and sustainability risk.
I hope my modification is satisfactory. Thank you again for your review and guidance!
